# Association between cardiometabolic disease and severe COVID-19: a nationwide case–control study of patients requiring invasive mechanical ventilation

Per Svensson ![ORCID] ,[1,2] Robin Hofmann,[1,2] Henrike Häbel,[3] Tomas Jernberg,[4] Per Nordberg[2,5]

For numbered affiliations see end of article.

**Correspondence to**
Dr Per Svensson;
Per.Svensson@ki.se

## ABSTRACT

**Aims** The risks associated with diabetes, obesity and hypertension for severe COVID-19 may be confounded and differ by sociodemographic background. We assessed the risks associated with cardiometabolic factors for severe COVID-19 when accounting for socioeconomic factors and in subgroups by age, sex and region of birth.

**Methods and results** In this nationwide case–control study, 1.086 patients admitted to intensive care with COVID-19 requiring mechanical ventilation (cases), and 10.860 population-based controls matched for age, sex and district of residency were included from mandatory national registries. ORs with 95% CIs for associations between severe COVID-19 and exposures with adjustment for confounders were estimated using logistic regression. The median age was 62 years (IQR 52–70), and 3003 (24.9%) were women. Type 2 diabetes (OR, 2.3 (95% CI 1.9 to 2.7)), hypertension (OR, 1.7 (95% CI 1.5 to 2.0)), obesity (OR, 3.1 (95% CI 2.4 to 4.0)) and chronic kidney disease (OR, 2.5 (95% CI 1.7 to 3.7)) were all associated with severe COVID-19. In the younger subgroup (below 57 years), ORs were significantly higher for all cardiometabolic risk factors. The risk associated with type 2 diabetes was higher in women (p=0.001) and in patients with a region of birth outside European Union(EU) (p=0.004).

**Conclusion** Diabetes, obesity and hypertension were all independently associated with severe COVID-19 with stronger associations in the younger population. Type 2 diabetes implied a greater risk among women and in non-EU immigrants. These findings, originating from high-quality Swedish registries, may be important to direct preventive measures such as vaccination to susceptible patient groups.

**Trial registration number** Clinicaltrial.gov (NCT04426084).

## INTRODUCTION

Observational data suggest both higher prevalence and more severe course of COVID-19 among individuals with diabetes,[1 2] obesity[3] and hypertension,[4–8] risk factors that are closely linked and cluster together in the metabolic syndrome.[9] As more clinical data

### Strength and limitations of this study

► In contrast to many previous reports, this study accounts for severity of COVID-19 and provides a homogeneous study group, by only including intubated patients at the intensive care with the highest risk of death. By inclusion of virtually all such cases nationwide, in a country with a tax-financed healthcare and a serious epidemic during the study period, the study provides a large sample size with adequate power and high external validity.

► This study compared patients with severe COVID-19 with matched controls in the underlying population and used 10 population-based, age, sex and district of residence-matched controls per each case. Therefore, the study can provide estimates of relative risk for severe COVID-19 in the population for the studied risk factors.

► Socioeconomic factors have been crucial in how the pandemic has impacted on different groups in the society but are also closely linked with obesity, diabetes and cardiovascular disease. By matching for district of residence and adjusting for individual level data on several socioeconomic variables, this study confirms previous studies and also provides novel evidence that diabetes, obesity and hypertension are independently associated with severe COVID-19 and how the relative importance of risk factors differs by age, sex and region of birth.

► The data on exposures are from high-quality national registries, and the patient cohort is the most complete Swedish cohort on severe COVID-19 published to date. The findings may be important to direct preventive measures such as vaccination to susceptible patient groups.

► A possible limitation is that the outcome did not include patients where admission to intensive care and/or mechanical ventilation was not considered appropriate and those who died before intensive care, another limitation is the lack of information on smoking status.

are emerging, new determinants of both COVID-19 and severity of disease are being discussed, such as coagulation disorders[10] and socioeconomic factors,[11] but to this

point, the underlying mechanisms that link cardiometabolic disease with severe COVID-19 remain unclear.[12]

In order to advance the knowledge on risk factors, several aspects are crucial to put evidence into perspective: First, the spectrum of disease severity needs to be addressed as the clinical presentation of infected patients can range from asymptomatic, to severe with high risk of fatal outcome. As the risks of being infected may differ from the risk of becoming severely ill once infected, there is a need for studies that focus on risk factors associated with a severe disease. Second, as socioeconomic and cultural factors are closely linked to type 2 diabetes, obesity and cardiovascular disease (CVD), these need to be accounted for in such analyses. And, most importantly, prevalent cases need to be compared with controls to reliably assess the magnitude of the major risk factors in the underlying population.[13] Currently, only a few population-based studies have investigated the association between cardiometabolic risk factors and COVID-19 death.[1 2 14] To our knowledge, no study has investigated whether cardiometabolic risk factors are independently associated with severe COVID-19 requiring intensive care, when controlled for age, sex, sociodemographic factors and immigrant background using matched population-based controls. In addition, it is unknown whether the impact of these risk factors is attenuated by age, sex and sociodemographic factors.

Sweden has been hit hard by the COVID-19 epidemic but in contrast to most other countries, did not employ a strict lockdown policy. To continuously evaluate the situation, strong governmental efforts were enforced on national healthcare registries for data reporting.

In this study, we present a comprehensive Swedish sample originating from mandatory, high-quality national registries with the aim to investigate whether cardiometabolic risk factors are associated with severe COVID-19 in patients treated at the intensive care unit (ICU) with invasive mechanical ventilation.

## METHODS
### Study design and ethics
This nationwide case–control study was based on data from the Swedish Intensive Care Registry (SIR) on patients (cases) with severe COVID-19 admitted to the ICU requiring invasive mechanical ventilation between 1 March and 11 May 2020. For each case, 10 controls were randomly selected from the Swedish Population Register and matched by age, sex and district of residence (corresponding to part of municipality). The study database was merged with multiple mandatory Swedish national registries at Statistics Sweden and the National Board of Health and Welfare, which are further described in online supplemental methods, using each individual's unique personal identification number.[15] The study used already collected, pseudonymised data and involved minimal infringement of personal integrity.

### Patient and public involvement
Patients and the public were not involved in the design or conduct of our research.

### National registries and data collection
Severe COVID-19 was defined as laboratory-confirmed COVID-19 infection in individuals treated at theICU with mechanical ventilation. These cases were reported to SIR,[16] which is a national register with about 95% coverage of all ICU admissions in Sweden and is used to identify eligible patients. The National Patient Register[17] was used to collect primary or secondary diagnoses from previous hospital admissions and outpatients' visits coded according to the International Classification of Diseases (ICD) V.10 within the 15 years preceding the admission. The Prescribed Drugs Register contains information on all dispensed drugs according to the Anatomical Therapeutic Chemical Classification. We collected individual data on dispensed drugs prescribed and claimed within 12 months before the study period. The longitudinal integrated database for health insurance and labour market studies is managed by Statistics Sweden and includes annual measurements on several socioeconomic and sociodemographic variables, including income, education and country of birth.[18]

### Definition of exposures
Exposures were a history of cardiometabolic or relevant chronic disease based on diagnoses in the National Patient Register within 15 years preceding the admission or prescribed drugs within the preceding 12 months. Hypertension (defined as previous diagnosis of ICD I10 or prescription of antihypertensive drugs as described previously,[19] hyperlipidaemia (ICD E78 or prescription of lipid lowering drugs), diabetes mellitus type 2 (ICD E11 or prescription of antidiabetic drugs), diabetes mellitus type 1 (ICD E10), obesity (ICD E66), heart failure (ICD I50.1, I50.9), atrial fibrillation (ICD I48), venous thromboembolism (ICD I26, I80), asthma (ICD J45), chronic obstructive pulmonary disease (ICD J44), chronic kidney disease (ICD N18), malignancy (ICD C, D40–48), rheumatoid arthritis (ICD M05, M06), systemic inflammatory disease (ICD M30–M36) and inflammatory bowel disease (ICD K50, K51) were included. A history of CVD was defined as a record of either myocardial infarction (MI) (ICD I21, I22), ischaemic heart disease (ICD I25), ischaemic stroke (ICD I63) or peripheral vascular disease (ICD I70–I73), in the Swedish Patient Register (online supplemental eTable1).

### Definition of covariates and variables for subgroup analyses
Level of education was categorised as ≤9 years (reference), 10–12 years and >12 years based on the highest educational level attained during the year before admission. Region of birth was categorised as a country of birth within European Union (EU)15 (Austria, Belgium, Denmark, Finland, France, Germany, Greece, Ireland, Italy, Luxembourg, Netherlands, Portugal, Spain, Sweden, UK) and/or the Nordic

Countries (Denmark, Finland, Iceland, Norway, Sweden) or having a country of birth outside this region. Marital status during the year before index date was categorised as married or not married that included unmarried, divorced and widowed. Subgroup analyses were performed for region of birth, sex (male/female) and age tertiles.

## Outcome

The outcome was defined as an ICU admission due to COVID-19 (with a laboratory-confirmed SARS-CoV-2 infection), registered in SIR, with at least one episode of invasive mechanical ventilation during the ICU stay. All eligible patients during the study period between 1 March and 11 May 2020 were included as cases in the study. In a sensitivity analysis, the outcome was defined as any ICU admission due to COVID-19 (with a laboratory confirmed SARS-CoV-2 infection), registered in SIR during the study period.

## Statistical methods

Categorical variables are reported as frequencies and percentages, while continuous variables are reported as median and IQR. Missing data are reported in online supplemental eTable 2.ORs and 95% CI for the association between the different exposures and the outcome were calculated by means of logistic regression adjusted for age and sex (model 1). For all exposures, additional adjustments were made for sociodemographic and socioeconomic variables (marital status, region of birth and educational level) (model 2) and, finally, for all conditions listed in the paragraph exposures above that were used as covariates in a fully adjusted regression model (model 3) in order to analyse both total effects unconfounded of sociodemographic variables and direct effects in accordance to our perception of causal relationships as illustrated by the directed acyclic graphs in online supplemental figure S1. SEs were calculated using the robust sandwich estimator and the significance level was set at an alpha of 0.05. For a formal test of a significant difference between the ORs for different subgroups, likelihood ratio tests were conducted between a model with and without an interaction term between the indicator variable for the subgroup and the risk factor. For these tests, the robust sandwich estimator was not used in the underlying logistic regression models.

Statistical analyses were performed using Stata V.16.0 (StataCorp, College Station, Texas, USA).

## RESULTS

During the study period between 1 March and 11 May 2020, a total of 1417 patients were admitted to an ICU in Sweden due to COVID-19 out of which 1086 required treatment with invasive mechanical ventilation (cases). For each case, 10 matched control subjects were randomly selected, rendering a total of 10 860 control subjects. The study population selection procedure and reasons for exclusions are described in online supplemental figure S2.

**Table 1** Baseline characteristics of the study population

| | COVID-19 (n=1086) | Control subjects (n=10 860) |
|---|---|---|
| Age, median (IQR), years | 62.0 (52.0–70.0) | 62.0 (52.0–70.0) |
| **Sex** | | |
| Male, number (%) | 813 (74.9) | 8130 (74.9) |
| **Sociodemographics** number (%) | | |
| Education (years) | | |
| ≤9 | 280 (26.8) | 2144 (20.1) |
| 10–12 | 466 (44.6) | 4805 (45.1) |
| ≥12 | 300 (28.7) | 3712 (34.8) |
| Marital status | | |
| Widow | 41 (3.8) | 405 (3.7) |
| Married | 632 (58.2) | 5641 (51.9) |
| Single | 218 (20.1) | 2802 (25.8) |
| Separated | 195 (18.0) | 2012 (18.5) |
| Region of birth | | |
| EU 15* and/or Nordics | 596 (55.1) | 8411 (77.5) |
| **Medical history** number (%) | | |
| Type 1 diabetes | 9 (0.8) | 39 (0.4) |
| Type 2 diabetes | 276 (25.4) | 1255 (11.6) |
| Obesity | 99 (9.1) | 328 (3.0) |
| Hypertension | 547 (50.4) | 4258 (39.2) |
| Hyperlipidaemia | 292 (33.0) | 2100 (28.6) |
| Chronic kidney disease | 40 (3.7) | 146 (1.3) |
| Cardiovascular disease | 105 (9.7) | 992 (9.1) |
| Myocardial infarction | 55 (5.1) | 559 (5.1) |
| Ischaemic stroke | 29 (2.7) | 274 (2.5) |
| Peripheral artery disease | 24 (2.2) | 249 (2.3) |
| Heart failure | 40 (3.7) | 329 (3.0) |
| Atrial fibrillation | 65 (6.0) | 589 (5.4) |
| Deep vein thrombosis | 40 (3.7) | 208 (1.9) |
| Pulmonary embolism | 13 (1.2) | 103 (0.9) |
| Chronic obstructive pulmonary disease | 32 (2.9) | 237 (2.2) |
| Asthma | 100 (9.2) | 376 (3.5) |
| Malignancy | 158 (14.5) | 1740 (16.0) |
| Rheumatoid arthritis | 17 (1.6) | 96 (0.9) |
| Systemic inflammatory disease | 33 (3.0) | 129 (1.2) |
| Inflammatory bowel disease | 17 (1.6) | 159 (1.5) |

Characteristics of patients with COVID-19 requiring mechanical ventilation and control subjects.
*EU 15 comprises of Austria, Belgium, Denmark, Finland, France, Germany, Greece, Ireland, Italy, Luxembourg, Netherlands, Portugal, Spain, Sweden, UK. The Nordic countries include Denmark, Finland, Iceland, Norway and Sweden.

## Patient characteristics

The median age was 62 (IQR 52–70) years and 75% were men. Baseline characteristics are summarised in table 1. Patients were less likely to have a postsecondary education and more likely to be married compared with the control group. Furthermore, patients were more likely to have

**Table 2** Pharmacological treatments of the study population

| | Covid-19 (n=1086) | Control subjects (n=10 860) |
|---|---|---|
| **Treatments** number (%) | | |
| *Antihypertensive treatments* | | |
| ACE inhibitors | 168 (15.5) | 1310 (12.1) |
| ARBs | 218 (20.1) | 1694 (15.6) |
| Calcium-channel blockers | 239 (22.0) | 1648 (15.2) |
| Beta-blockers | 222 (20.4) | 1849 (17.0) |
| Diuretics | 51 (4.7) | 522 (4.8) |
| *Antidiabetic treatments* | | |
| Any antidiabetics | 240 (22.1) | 1144 (10.5) |
| Insulins | 80 (7.4) | 391 (3.6) |
| Biguanides | 200 (18.4) | 855 (7.9) |
| Sulfonylureas | 28 (2.6) | 93 (0.9) |
| Glitazons | 6 (0.6) | 10 (0.1) |
| DPP-4 inhibitors | 44 (4.1) | 210 (1.9) |
| GLP-1 RAs | 37 (3.4) | 184 (1.7) |
| SGLT-2 inhibitors | 46 (4.2) | 184 (1.7) |
| Meglitinides | 4 (0.4) | 34 (0.3) |
| Statins | 288 (26.5) | 2242 (20.6) |
| Aspirin | 136 (12.5) | 1103 (10.2) |
| Other antiplatelet drugs | 20 (1.8) | 234 (2.2) |
| Warfarin | 17 (1.6) | 150 (1.4) |
| NOAC | 45 (4.1) | 474 (4.4) |

ACE, angiotensin converting enzyme; ARBs, angiotensin receptor blockers; DDP-4, dipeptidyl peptidase-4; NOAC, new oral anticoagulants; GLP-1 RAs, glucagon-like peptide-1 receptor agonists; SGLT2, sodium-glucose cotransporter-2.

history of migration with more patients having a region of birth outside EU 15 and the Nordic countries. Comorbid conditions were more common among patients with COVID-19 receiving mechanical ventilation compared with the control group. In particular, cardiometabolic risk factors were over-represented with more patients having not only a history of hypertension, hyperlipidaemia, diabetes mellitus, obesity and chronic kidney disease but also venous thromboembolic disease, asthma and systemic inflammatory diseases, which were more common. Due to more comorbid conditions, patients had correspondingly more pharmacological treatments (table 2). All antihypertensive treatments were more common among patients compared with control subjects as were all antidiabetic treatments except meglitinides.

## Comparison of risk factors and treatments between patient with severe COVID-19 and controls

In the multivariable logistic regression models presented in table 3, both type 2 diabetes (OR, 2.7 (95% CI 2.3 to 3.2)), hypertension (OR, 1.8 (95% CI 1.5 to 2.0)), hyperlipidaemia (OR, 1.4 (95% CI 1.2 to 1.6)), obesity (OR,

3.2 (95% CI 2.6 to 4.1)) and chronic kidney disease (OR, 2.8 (95% CI 2.0 to 4.0)) were associated with COVID-19 receiving mechanical ventilation. All associations remained significant after adjustment for possible socioeconomic confounders. In the fully adjusted model, all cardiometabolic risk factors except hyperlipidaemia were associated with the outcome indicating direct and additive effects for these risk factors (figure 1). In addition, we observed associations between a history of venous thromboembolic disease (OR, 1.7 (95% CI 1.3 to 2.4)), asthma (OR, 2.8 (95% CI 2.3 to 3.6)), rheumatoid arthritis (OR, 1.8 (95% CI 1.1 to 3.0)) as well as systemic inflammatory disease (OR, 2.6 (95% CI 1.8 to 3.9)) and severe COVID-19. In contrast, neither CVD, heart failure, atrial fibrillation, malignancy, chronic obstructive pulmonary disease nor inflammatory bowel disease was associated with the outcome.

In the logistic regression analysis adjusted for age and sex using no treatment as reference, all types of antihypertensive treatment, except diuretics and all types of antidiabetic treatments, except meglitinides, were associated with severe COVID-19 (table 4). However, when adjusting for all comorbidities in the fully adjusted model, calcium-channel blockers, biguanides and glitazones were the only treatments that remained associated with the outcome.

## Subgroup analysis

Baseline characteristics by subgroups of age, sex and region of birth are summarised in online supplemental tables 3 to 5, respectively. A regression analysis for the cardiometabolic risk factors is presented in table 5 by subgroups of age (tertiles), sex and region of birth. In the younger subgroup (aged 21–56 years), ORs were significantly higher for hypertension (OR, 2.6 (95% CI 2.0 to 3.3)), type 2 diabetes (OR, 4.5 (95% CI 3.3 to 6.2)), obesity (OR, 7.6 (95% CI 5.3 to 11.0)), chronic kidney disease (OR, 7.7 (95% CI 3.4 to 17.5)) (p=0.010), venous thromboembolic disease (OR, 3.9 (95% CI 2.0 to 7.6)), asthma (OR, 4.9 (95% CI 3.3 to 7.3)), systemic inflammatory disease (OR, 6.7 (95% CI 3.0 to 14.9)) and heart failure (OR, 5.4 (95% CI 2.4 to 12.2)) as illustrated in figure 2. In women, the ORs for type 2 diabetes and asthma were significantly higher as compared with men (figure 3). Among patients with a region of birth outside EU 15, diabetes had a stronger association with severe COVID-19 compared with patients with a region of birth within EU 15, whereas a trend towards the opposite was observed for obesity (figure 4).

## Sensitivity analysis

In a sensitivity analysis, we report any COVID-19-related ICU admission (with or without mechanical ventilation) as the outcome (online supplemental table 6), which was found in a total of 1417 patients. Associations with cardiometabolic risk factors were similar but positive associations were also observed with heart failure (OR, 1.6 (95% CI 1.2 to 2.1)), atrial fibrillation (OR, 1.5 (95%

**Table 3** ORs for COVID-19 requiring mechanical ventilation by cardiometabolic factors and other comorbidities

| Risk factors | Adjusted for age and sex | | | Adjusted model 2* | | | Adjusted model 3 † | | |
|---|---|---|---|---|---|---|---|---|---|
| | OR | 95% CI | P value | OR | 95% CI | P value | OR | 95% CI | P value |
| Type 1 diabetes | 2.32 | 1.12 to 4.81 | 0.023 | 3.13 | 1.52 to 6.42 | 0.002 | 2.56 | 1.25 to 5.24 | 0.010 |
| Type 2 diabetes | 2.73 | 2.33 to 3.20 | <0.001 | 2.25 | 1.90 to 2.65 | <0.001 | 1.81 | 1.49 to 2.19 | <0.001 |
| Obesity | 3.23 | 2.56 to 4.08 | <0.001 | 3.13 | 2.43 to 4.02 | <0.001 | 2.03 | 1.55 to 2.65 | <0.001 |
| Hypertension | 1.76 | 1.52 to 2.05 | <0.001 | 1.73 | 1.48 to 2.01 | <0.001 | 1.26 | 1.05 to −1.51 | <0.013 |
| Hyperlipidaemia | 1.35 | 1.15 to 1.58 | <0.001 | 1.22 | 1.03 to 1.43 | <0.018 | 0.90 | 0.75 to 1.09 | 0.286 |
| CKD | 2.83 | 1.97 to 4.05 | <0.001 | 2.51 | 1.69 to 3.70 | <0.001 | 1.84 | 1.21 to 2.82 | 0.005 |
| CVD | 1.07 | 0.86 to 1.33 | 0.554 | 1.03 | 0.82 to 1.29 | 0.789 | 0.75 | 0.58 to 0.96 | 0.022 |
| Heart failure | 1.23 | 0.87 to 1.73 | 0.236 | 1.13 | 0.79 to 1.62 | 0.48 | 0.78 | 0.51 to 1.19 | 0.253 |
| Atrial fibrillation | 1.11 | 0.85 to 1.46 | 0.43 | 1.24 | 0.93 to 1.64 | 0.136 | 1.04 | 0.75 to 1.43 | 0.819 |
| VTE | 1.74 | 1.27 to 2.39 | 0.001 | 1.90 | 1.37 to 2.62 | <0.001 | 1.65 | 1.18 to 2.31 | 0.004 |
| COPD | 1.37 | 0.94 to 1.99 | 0.105 | 1.34 | 0.91 to 1.97 | 0.133 | 0.87 | 0.56 to 1.36 | 0.552 |
| Asthma | 2.84 | 2.26 to 3.58 | <0.001 | 2.78 | 2.18 to 3.53 | <0.001 | 2.25 | 1.74 to 2.90 | <0.001 |
| Malignancy | 0.89 | 0.74 to 1.06 | 0.195 | 0.97 | 0.80 to 1.17 | 0.725 | 0.85 | 0.70 to 1.04 | 0.103 |
| Rheumatoid arthritis | 1.79 | 1.06 to 3.00 | 0.029 | 1.86 | 1.11 to 3.12 | 0.019 | 1.27 | 0.72 to 2.23 | 0.407 |
| Systemic inflammatory disease | 2.65 | 1.79 to 3.92 | <0.001 | 2.57 | 1.71 to 3.86 | <0.001 | 1.96 | 1.28 to 2.99 | 0.002 |
| Inflammatory bowel disease | 1.07 | 0.65 to 1.77 | 0.792 | 1.21 | 0.72 to 2.03 | 0.479 | 0.94 | 0.54 to 1.64 | 0.839 |

*Adjusted for age, sex, educational level, marital status and region of birth.
†Adjusted for age, sex, educational level, marital status and region of birth and all diagnoses in table 3.
CKD, chronic kidney disease; COPD, chronic obstructive pulmonary disease; CVD, cardiovascular disease; VTE, venous thromboembolism.

CI 1.2 to 1.9)) and chronic obstructive pulmonary disease (OR, 1.9 (95% CI 1.4 to 2.5)).

## DISCUSSION

In the present nationwide case–control study, assessing the risk for severe COVID-19 with need for mechanical

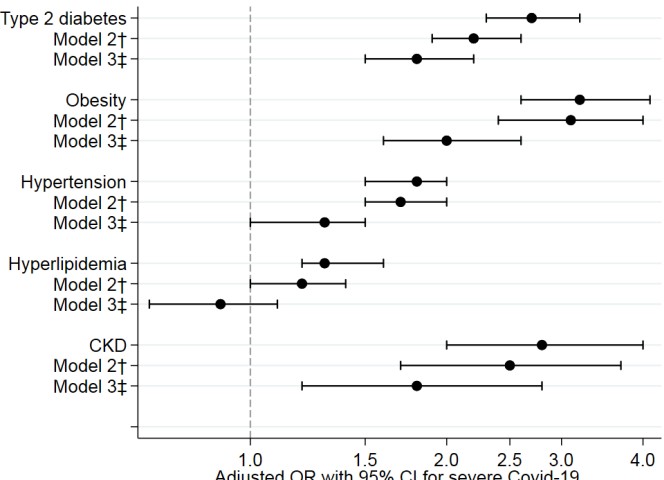

**Figure 1** Associations of cardiometabolic risk factors with severe COVID-19 (adjusted ORs with 95% CIs). †Adjusted for age, sex, educational level, marital status and region of birth. ‡Adjusted for age, sex, educational level, marital status, region of birth and all diagnoses in table 3.

ventilation at the ICU, we found that the cardiometabolic risk factors, such as diabetes, obesity and hypertension, were strongly and independently associated with severe infection also when accounting for socioeconomic factors. Furthermore, we found higher risks associated with all cardiometabolic risk factors among younger patients, whereas diabetes was more important in women and in those with an immigrant background. These findings, originating from high-quality national registries in Sweden, which has been experiencing a serious epidemic, confirm and extend findings from previous studies and may be important to identify susceptible patient groups requiring extra precautions and prioritised for vaccination.

Cardiometabolic risk factors were early linked with a severe COVID-19 in case series[20] and uncontrolled studies,[21] later in studies using population-based control subjects[5] and nationwide studies based on data from electronic health records.[1 2 14] Here, we confirm these findings and extend them in a well-controlled design and can reinforce that diabetes, obesity and hypertension—risk factors that are closely linked and often cluster together in the metabolic syndrome[22]—all have strong and independent direct associations with the outcome. By adjusting for sociodemographic factors collected on individual level, we can also show that obesity and other components of the metabolic syndrome—factors that are closely linked with lower socioeconomic status[23]—act on

**Table 4** ORs for COVID-19 requiring mechanical ventilation by pharmacological treatments

| Treatments | Adjusted for age and sex | | | Adjusted model 2* | | | Adjusted model 3 † | | |
|---|---|---|---|---|---|---|---|---|---|
| | OR | 95% CI | P value | OR | 95% CI | P value | OR | 95% CI | P value |
| *Antihypertensive treatments* | | | | | | | | | |
| ACE inhibitors | 1.35 | 1.13 to 1.62 | 0.001 | 1.35 | 1.12 to 1.62 | 0.797 | 0.99 | 0.81 to 1.22 | 0.931 |
| ARBs | 1.39 | 1.18 to 1.64 | <0.001 | 1.47 | 1.24 to 1.75 | <0.001 | 1.07 | 0.88 to 1.30 | 0.474 |
| CCBs | 1.64 | 1.39 to 1.93 | <0.001 | 1.64 | 1.39 to 1.93 | <0.001 | 1.25 | 1.03 to 1.52 | 0.025 |
| Beta-blockers | 1.28 | 1.08 to 1.51 | 0.004 | 1.27 | 1.07 to 1.51 | 0.007 | 0.90 | 0.73 to 1.11 | 0.345 |
| Diuretics | 0.98 | 0.72 to 1.32 | 0.870 | 1.00 | 0.74 to 1.36 | 0.996 | 0.74 | 0.53 to 1.03 | 0.078 |
| *Antidiabetic treatments* | | | | | | | | | |
| Insulins | 2.15 | 1.67 to 2.77 | <0.001 | 1.90 | 1.46 to 2.47 | <0.001 | 0.85 | 0.62 to 1.16 | 0.305 |
| Biguanides | 2.72 | 2.28 to 3.24 | <0.001 | 2.26 | 1.88 to 2.72 | <0.001 | 1.40 | 1.01 to 1.94 | 0.044 |
| Sulfonylureas | 3.08 | 2.01 to 4.73 | <0.001 | 2.13 | 1.33 to 3.41 | 0.002 | 1.17 | 0.72 to 1.91 | 0.530 |
| Glitazons | 6.03 | 2.19 to 16.6 | 0.001 | 5.46 | 1.98 to 15.0 | 0.001 | 2.86 | 1.04 to 7.85 | 0.042 |
| DPP-4 inhibitors | 2.16 | 1.54 to 3.01 | <0.001 | 1.77 | 1.25 to 2.50 | 0.001 | 0.91 | 0.62 to 1.33 | 0.632 |
| GLP-1 RA | 2.79 | 1.92 to 4.03 | <0.001 | 2.71 | 1.85 to 3.97 | <0.001 | 1.24 | 0.82 to 1.87 | 0.312 |
| SGLT-2 inhibitors | 2.58 | 1.85 to 3.59 | <0.001 | 2.30 | 1.63 to 3.25 | <0.001 | 1.20 | 0.82 to 1.74 | 0.346 |
| Meglitinides | 1.18 | 0.42 to 3.33 | 0.383 | 1.03 | 0.36 to 2.96 | 0.953 | 0.55 | 0.19 to 1.61 | 0.276 |
| Statins | 1.44 | 1.23 to 1.69 | <0.001 | 1.32 | 1.12 to 1.55 | 0.001 | 0.84 | 0.63 to 1.13 | 0.247 |
| Aspirin | 1.29 | 1.05 to 1.57 | 0.013 | 1.14 | 0.93 to 1.41 | 0.200 | 0.97 | 0.75 to 1.24 | 0.791 |
| Warfarin | 1.14 | 0.68 to 1.90 | 0.622 | 1.29 | 0.76 to 2.17 | 0.347 | 0.96 | 0.52 to 1.76 | 0.894 |
| NOAC | 0.95 | 0.69 to 1.30 | 0.730 | 1.04 | 0.75 to 1.44 | 0.806 | 0.70 | 0.46 to 1.06 | 0.093 |

*Adjusted for age, sex, educational level, marital status and region of birth.
†Adjusted for age, sex, educational level, marital status and region of birth and all diagnoses in table 3
ACE, angiotensin converting enzyme; ARBs, angiotensin receptor blockers; CCBs, calcium-channel blockers; DPP-4, dipeptidyl peptidase-4; NOAC, new oral anticoagulants; GLP-1 RA, glucagon-like peptide-1 receptor agonists; SGLT2, sodium-glucose cotransporter-2.

COVID-19 independent of sociodemography. The effect of diabetes was even stronger in the population with immigrant background.

Initial studies on hypertension as a risk factor for severe disease[24] may have been confounded by age and until now, there has been limited evidence of hypertension being an independent risk factor.[25] In a large nationwide study from the UK on risk factors for death in COVID-19,[1] hypertension was positively associated with the outcome when adjusted for age and sex, but it was no longer a risk factor when other comorbid conditions were accounted in the fully adjusted model. In another large study in patient with type 2 diabetes, antihypertensive treatment was independently associated with COVID-19-related mortality.[14] Here, we can report that hypertension is not only a risk factor independent of age and other related conditions but also that risk factor patterns differ by age, sex and region of birth. In the younger subgroup (age below 57 years), all cardiometabolic factors had an even stronger association with severe COVID-19. In women and individuals born outside EU 15, diabetes had the strongest association with the outcome. All components of metabolic syndrome are associated with endothelial dysfunction[26] and low-grade inflammation.[27] Hypertension is also linked with a dysregulated immune system,[28] including endothelial mechanisms,[29] and is causally associated with increased lymphocyte count.[30] As emerging evidence suggests that endothelial inflammation is involved in serious manifestations of COVID-19,[31] it is possible that a common mechanism linking cardiometabolic risk factors with severe COVID-19 is mediated through endothelial and microcirculatory dysfunction. Our findings suggest that these mechanisms are even more important at a younger age. Therefore, a more detailed metabolic phenotyping that includes biomarkers of subclinical inflammation, and insulin resistance may be important to identify younger patients at the highest risk, and further studies are warranted in this field.[3]

We identified asthma, previous thromboembolic disease, rheumatoid arthritis and systemic inflammatory disease as additional chronic diseases with increased risk for a severe course of COVID-19. This is new and important information as patients with chronic inflammatory conditions may also be more susceptible to the proinflammatory pathways of the infection that involves diffuse endothelial inflammation and systemic impaired microcirculation leading to multiorgan dysfunction.[27]

The association between socioeconomic factors and CVD is well established,[23] and socioeconomic factors have also been important during the COVID-19 pandemic.[32 33]

**Table 5** COVID-19 risk factors by subgroups of age, sex and region of birth

| | Type 2 diabetes | Obesity | Hypertension | CKD | CVD | VTE | Asthma | SID | Heart failure |
|---|---|---|---|---|---|---|---|---|---|
| **Age** | | | | | | | | | |
| 21–56 years | 4.5 (3.3–6.2) | 7.6 (5.3–11.0) | 2.6 (2.0–3.3) | 7.7 (3.4–17.5) | 1.6 (0.8–3.3) | 3.9 (2.0–7.6) | 4.9 (3.3–7.3) | 6.7 (3.0–14.9) | 5.4 (2.4–12.2) |
| 57–67 years | 2.6 (2.0–3.4)* | 3.7 (2.5–5.4)* | 1.7 (1.3–2.1)* | 3.2 (1.6–6.4) | 1.4 (1.0–2.0) | 1.9 (1.1–3.2) | 3.7 (2.7–5.1) | 2.5 (1.1–5.5) | 1.5 (0.8–2.6)* |
| 68–87 years | 1.4 (1.1–1.9)* | 2.2 (1.3–3.7)* | 1.3 (1.0–1.7)* | 1.5 (0.8–2.7)* | 0.9 (0.7–1.2) | 1.4 (0.8–2.3)* | 2.3 (1.5–3.5)* | 2.0 (1.2–3.7)* | 0.8 (0.5–1.4)* |
| **Sex** | | | | | | | | | |
| Male | 2.0 (1.7–2.4) | 3.1 (2.3–4.3) | 1.7 (1.4–2.0) | 2.2 (1.4–3.4) | 1.0 (0.8–1.3) | 1.8 (1.3–2.6) | 2.3 (1.7–3.1) | 2.1 (1.1–3.9) | 1.0 (0.7–1.5) |
| Female | 3.5 (2.5–5.0) | 3.2 (2.0–5.0) | 1.8 (1.3–2.5) | 4.2 (1.9–9.3) | 1.2 (0.7–2.2) | 2.2 (1.1–4.2) | 3.9 (2.6–5.8) | 3.1 (1.8–5.3) | 2.0 (0.9–4.4) |
| P value | 0.001 | 0.922 | 0.327 | 0.113 | 0.416 | 0.558 | 0.030 | 0.019 | 0.103 |
| **Region of birth** | | | | | | | | | |
| Outside EU15 | 3.3 (2.6–4.2) | 2.3 (1.6–3.4) | 1.8 (1.5–2.3) | 2.5 (1.4–4.6) | 1.4 (1.0–1.9) | 1.4 (0.8–2.4) | 3.2 (2.2–4.5) | 2.1 (1.1–4.0) | 1.3 (0.8–2.31) |
| EU 15/Nordic | 2.0 (1.6–2.6) | 3.7 (2.7–5.1) | 1.6 (1.3–1.9) | 2.6 (1.6–4.2) | 0.9 (0.7–1.2) | 2.0 (1.4–3.0) | 2.7 (1.9–3.7) | 2.8 (1.7–4.6) | 1.1 (0.7–1.7) |
| P value | 0.004 | 0.066 | 0.476 | 0.872 | 0.072 | 0.248 | 0.584 | 0.584 | 0.651 |

ORs for COVID-19 receiving mechanical ventilation compared with matched control subjects by tertiles of age, sex and region of birth (EU15).
P values denote likelihood-ratio tests between a model with and one without an interaction term between the indicator variable for the subgroup and the risk factor added to model 2. Age subgroups were compared with 21–56 years, p values for interaction are presented as *P value<0.05.
CKD, chronic kidney disease; CVD, cardiovascular disease (history of myocardial infarction, ischaemic stroke or peripheral arterial disease); SID, systemic inflammatory disease; VTE, venous thromboembolic disease.

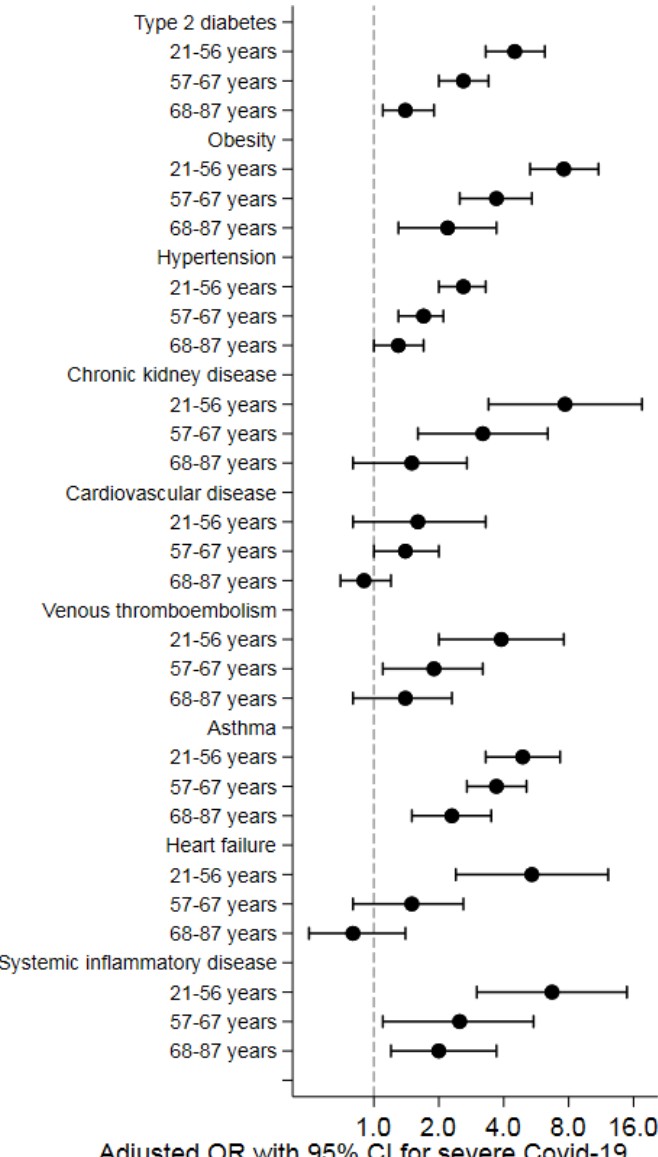

**Figure 2** Associations of cardiometabolic risk factors with severe COVID-19 by tertiles of age (adjusted ORs with 95% CIs).

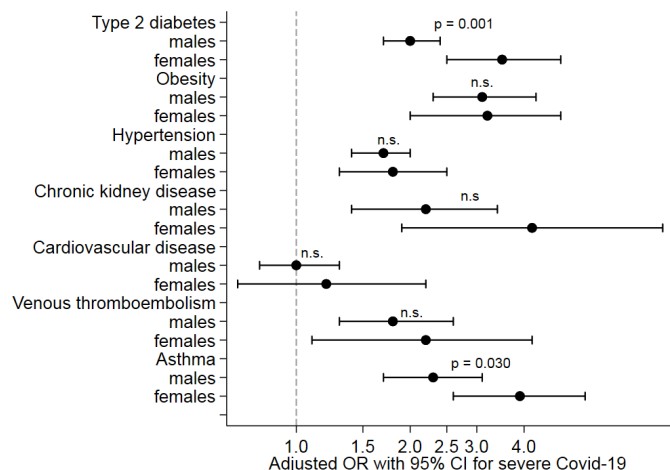

**Figure 3** Associations of cardiometabolic risk factors with severe COVID-19 by sex (adjusted ORs with 95% CIs). P values denote likelihood-ratio tests between a model with and one without an interaction term between the indicator variable for the subgroup and the risk factor added to model 2.

We, therefore, believe that matching for residency, which is linked with socioeconomic factors, as well as adjustments for individual level information on migration, level of education and marital status, is a crucial factor in our study design.

In comparison to several previous descriptive studies, we have a well-characterised, homogenous, nationwide population with severe disease, all needing mechanical ventilation at the ICU. To the best of our knowledge, this is the first study that includes virtually all cases with this type of severe disease nationwide together with a population-based matched control group. In addition, some previous studies include a heterogeneous mix of cases where other factors such as testing patterns in mild cases may have influenced overall results.[5] Consequently, the current study estimated relative risks in the population for developing severe COVID-19, which may differ from

the risk of obtaining an infection with a milder course of disease. Since Swedish healthcare is virtually fully tax funded, all acute treatments, including admission to the ICU with invasive mechanical ventilation, are based on medical decisions and do not involve private-economic considerations. Unequal access to healthcare is thereby reduced, increasing the validity of our results. In addition, vast resources were put to scale-up ICU resources due to the pandemic, which resulted in a nationwide surplus of ICU beds during the study period; however, there are some limitations with the present study. It is

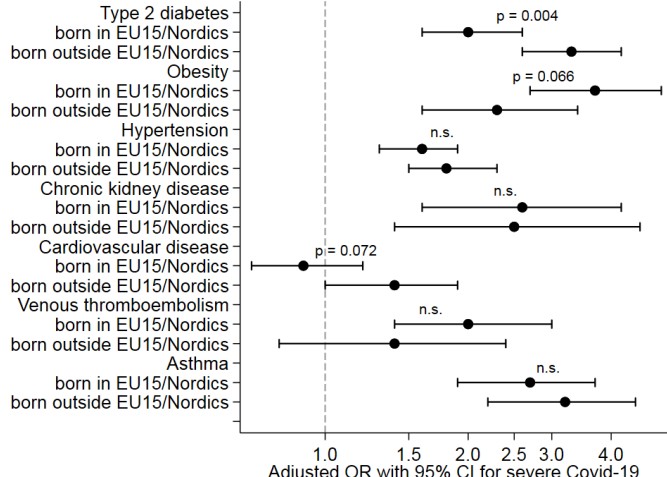

**Figure 4** Associations of cardiometabolic risk factors with severe COVID-19 by region of birth * (adjusted ORs with 95% CIs). *EU 15 comprises of Austria, Belgium, Denmark, Finland, France, Germany, Greece, Ireland, Italy, Luxembourg, Netherlands, Portugal, Spain, Sweden, UK. The Nordic countries include Denmark, Finland, Iceland, Norway and Sweden. P values denote likelihood-ratio tests between a model with and one without an interaction term between the indicator variable for the subgroup and the risk factor added to model 2.

possible that intensive care treatment with mechanical ventilation was not considered appropriate in some patients with multiple comorbidities or severe frailty, thus, our results may underestimate associations with severe COVID-19 for these conditions. Although the indication for early mechanical ventilation in severe COVID-19 has changed somewhat as the pandemic evolved, we think this will have limited impact on our results since we only included the first 2 months of the first wave in which the indication was rather stable. Furthermore, in terms of external validity, patient selection for intensive care treatment including invasive ventilation may differ between countries,[34] however, it is unlikely that this will affect the relative importance of risk factors. We did not have any information on smoking or concerning the rate of mild infection with COVID-19 among control subjects. Finally, the observational design of the study cannot exclude the potential of residual confounding and the results should be interpreted as such.

## CONCLUSIONS

Diabetes, obesity and hypertension were all independently associated with severe COVID-19 requiring mechanical ventilation at the ICU, with strongest associations in the younger population. Type 2 diabetes implied a greater risk among women and in those with immigrant background. These findings, originating from high-quality Swedish registries, may be important to direct preventive measures such as vaccination to susceptible patient groups.

**Author affiliations**
[1]Department of Clinical Science and Education, Södersjukhuset, Karolinska Institutet, Stockholm, Sweden
[2]Department of Cardiology, Södersjukhuset, Stockholm, Sweden
[3]Institute of Environmental Medicine, Karolinska Institutet, Stockholm, Sweden
[4]Department of Clinical Sciences, Danderyd University Hospital, Karolinska Institutet, Stockholm, Sweden
[5]Department of Medicine Solna, Centre for Resuscitation Science, Karolinska institutet, Stockholm, Sweden

**Acknowledgements** We thank Maria Ioanna Kotopouli, Karolinska Institutet, for expert statistical support.

**Contributors** PS had full access to all the data in the study and takes full responsibility for the integrity of the data and accuracy of the data analysis. Study concept and design: PS, PN, HH.Acquisition of data: PS. Analysis and interpretation of data: PS, RH, TJ, PN, HH. Drafting the manuscript: PS, RH, PN. Critical revision of the manuscript for important intellectual content: PS, RH, TJ, PN, HH. Statistical analysis: HH. Obtained funding: PS.

**Funding** This work was supported by grants from the Region Stockholm (ALF-project, grant number 2019-0100). RH was supported by Region Stockholm (clinical postdoctoral appointment, grant number K 2017-4577) and the Swedish Heart Lung foundation (grant number 20180187).

**Competing interests** None declared.

**Patient consent for publication** Not required.

**Ethics approval** The study, complied with the Declaration of Helsinki, was approved by the National Ethical Review Board (identification number 2020/124-31/4).

**Provenance and peer review** Not commissioned; externally peer reviewed.

**Data availability statement** Data are available upon reasonable request. The data underlying this article cannot be shared publicly due to the privacy of individuals that participated in the study. The data will be shared on reasonable request to the corresponding author.

**ORCID iD**
Per Svensson http://orcid.org/0000-0003-0372-6272

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
