## [Reviewer comments · BMJ Open]

ARTICLE DETAILS

TITLE (PROVISIONAL)	Association between cardiometabolic disease and severe Covid-19: a nationwide case-control study of patients requiring invasive mechanical ventilation
AUTHORS	Svensson, Per; Hofmann, Robin; Häbel, Henrike; Jernberg, Tomas; Nordberg, Per

VERSION 1 – REVIEW

REVIEWER	Norbert Stefan University Hospital of Tübingen, Germany
REVIEW RETURNED	29-Sep-2020

GENERAL COMMENTS	In this study the authors investigated the risks associated with cardiometabolic factors for severe Covid-19, when accounting for socioeconomic factors and in subgroups by age, sex and region of birth, in Sweden. For this they designed a nationwide case-control study, including 1.086 patients admitted to intensive care with Covid-19 requiring mechanical ventilation (cases) and 10.860 population-based controls matched for age, sex and district of residency, from mandatory national registries. They found that diabetes, obesity and hypertension were all independently associated with severe Covid-19, with stronger associations in the younger population. Comments: 1. As cardiometabolic risk factors are very important for a severe course of COVID-19 in younger people, the authors should also run subgroup analyses by tertiles or quartiles of age, if sufficient numbers of patients can be found in these subgroups.2. What may explain this higher risk for a severe course of COVID-19 in metabolically unhealthy and obese younger patients? Apparently information about smoking, which may have been more often found in younger people, was not available.3. Can the authors test whether younger metabolically unhealthy patients received less pharmacological treatment and whether this may partially explain the different findings in younger compared to older people?4. Most probably, a survival effect, in that older people with more severe cardiometabolic risk and higher prevalence of obesity, died before being intubated, is not very probable. To test this analysis of hospital admission as an outcome would be helpful.5. As the authors report very important data, they should better highlight in the discussion that, in addition to focusing on obesity and the diagnosis of diabetes, a more in depth metabolic phenotyping, including measurements of subclinical inflammation and insulin resistance (Nat Rev Endocrinol. 2020 Jul;16(7):341-342.), may also be critical for risk stratification of the course of COVID-19, particularly in younger people.
---

REVIEWER	Naomi Holman University of Glasgow, UK
REVIEW RETURNED	05-Nov-2020

GENERAL COMMENTS	This is a timely paper that explore the cardiometabolic risk factors associated with mechanical ventilation in the Swedish population during the initial period of the COVID-19 pandemic. It makes good use of data collected in national registries and I have concerns about the use of invasive mechanical ventilation as an outcome in this study. I understand the authors which to make a distinction between mild and severe COVID-19. However, the use of mechanical ventilation as an outcome is problematic. Early in the time period covered by the analysis early mechanical ventilation was advocated but this view changed as the pandemic unfolded and knowledge of COVID-19 developed. Using mechanical ventilation as an outcome does not address the outcomes of those where admission to ICU and/or mechanical ventilation were not considered appropriate (eg the very frail, those with extensive co-morbidities, etc) or those who died before accessing hospital/ICU care. The discussion identifies the universal nature of the health system in Sweden but does not discuss how decisions to allocate ICU beds and therefore facilities to ventilate may have been taken during the study period given the pressures that the health service was experiencing at the time. These factors mean that there are uncertainties around the outcome. An analysis of death following diagnosis of COVID-19, especially over the longer time period that has now elapsed since the original study, may provide more robust findings. The paper repeatedly states that no other paper has assessed the risk associated with cardiometabolic risk factors and severe COVID-19 in comparison to a control population with matched demographic characteristics. However, there are two very large scale studies of English populations which use administrative data to consider this question (https://pubmed.ncbi.nlm.nih.gov/32640463/ and https://pubmed.ncbi.nlm.nih.gov/32798472/). Both of these papers use death related to COVID-19 as the outcome and review cardiometabolic factors in the context of social demographic characteristics. This paper needs to be reviewed after considering these papers and be considered as a study confirming these associations in a Swedish population. Some minor points for consideration  • The dates of the study are not reported in the methods section. • The time period in which co-morbidities were identified is only specified for some conditions. • The methods section does not state how COVID-19 status was identified. • The methods section does not include any details of the information governance/data protection processes that facilitated this work. • The results section could be improved by quoting the specific odds ratios and confidence intervals rather that relying on looking up all values in the tables.
--

VERSION 1 – AUTHOR RESPONSE

Reviewer: 1

Comments to the Author

In this study the authors investigated the risks associated with cardiometabolic factors for severe Covid-19, when accounting for socioeconomic factors and in subgroups by age, sex and region of birth, in Sweden. For this they designed a nationwide case-control study, including 1.086 patients admitted to intensive care with Covid-19 requiring mechanical ventilation (cases) and 10.860 population-based controls matched for age, sex and district of residency, from mandatory national registries. They found that diabetes, obesity and hypertension were all independently associated with severe Covid-19, with stronger associations in the younger population.

Comments:

1. As cardiometabolic risk factors are very important for a severe course of COVID-19 in younger people, the authors should also run subgroup analyses by tertiles or quartiles of age, if sufficient numbers of patients can be found in these subgroups.

Author response: We agree that more detailed information regarding different age groups is of interest for the readers and relevant in this setting. Therefore, in the revised version of the manuscript, we now present how cardiometabolic risk factors and other comorbidities relate to the risk for severe COVID-19 in tertiles of age in table 3 and figure 2.

In addition, descriptive data for these age-groups are presented in supplementary eTable3a-c. This analysis clearly shows that cardiometabolic risk factors are relatively more important for severe Covid-19 in the younger age groups. It also shows that diagnoses such as heart failure and systemic inflammatory disease are more important at younger age.

We think this additional analysis has improved the paper and would like to thank the reviewer for this suggestion.

2. What may explain this higher risk for a severe course of COVID-19 in metabolically unhealthy and obese younger patients? Apparently information about smoking, which may have been more often found in younger people, was not available.

Author response: This is an intriguing question and we are to some extent discussing that endothelial dysfunction and low-grade inflammation possibly may be involved in the discussion section of the manuscript. The reviewer is correct in assuming that information on smoking status is unfortunately not available in the data. We have now stated this fact as a limitation in the revised manuscript.

3. Can the authors test whether younger metabolically unhealthy patients received less pharmacological treatment and whether this may partially explain the different findings in younger compared to older people?

Author response: We agree that this information would also be of interest, however we don't think it is possible to test in the current data. Although our data do include pharmacological treatment, the sources of our data do not include other detailed information on metabolic phenotypes (such as waist

circumference) which would make such an analysis meaningful. However we don't see that this can explain our results, as we use pick-up of antidiabetic

drugs to identify patients with T2DM, the stronger association with T2DM that we observe in younger patients is to a large extent dependent on that there were more pick-ups of antidiabetic drugs (not less) in patients with the outcome.

4. Most probably, a survival effect, in that older people with more severe cardiometabolic risk and higher prevalence of obesity, died before being intubated, is not very probable. To test this analysis of hospital admission as an outcome would be helpful.

Author response: We agree that an analysis of all hospital admission with Covid-19 nationwide would provide additional info. However, to have a more homogenous population with severe COVID-19, we used the Swedish intensive care registry to identify patients, and thus only included patients admitted to intensive care. Unfortunately, we do not have information on other hospital admission due to COVID-19. There are both advantages and limitations with this patient selection, which has been further clarified in the section of strengths and limitations. We have performed an additional sensitivity analysis using all admissions to ICU (with or without mechanical ventilation) as an alternative outcome which is presented as supplementary etable6 in the revised version. In this analysis, the associations with cardiometabolic disease were similar.

5. As the authors report very important data, they should better highlight in the discussion that, in addition to focusing on obesity and the diagnosis of diabetes, a more in depth metabolic phenotyping, including measurements of subclinical inflammation and insulin resistance (Nat Rev Endocrinol. 2020 Jul;16(7):341-342.), may also be critical for risk stratification of the course of COVID-19, particularly in younger people.

Author response: Thank you for this comment. We agree that this information is of interest and we have added this important reference to the discussion where we think it fits very well.

Reviewer: 2

Comments to the Author

This is a timely paper that explore the cardiometabolic risk factors associated with mechanical ventilation in the Swedish population during the initial period of the COVID-19 pandemic. It makes good use of data collected in national registries and I have concerns about the use of invasive mechanical ventilation as an outcome in this study.

I understand the authors which to make a distinction between mild and severe COVID-19. However, the use of mechanical ventilation as an outcome is problematic. Early in the time period covered by the analysis early mechanical ventilation was advocated but this view changed as the pandemic unfolded and knowledge of COVID-19 developed. Using mechanical ventilation as an outcome does not address the outcomes of those where admission to ICU and/or mechanical ventilation were not considered appropriate (eg the very frail, those with extensive co-morbidities, etc) or those who died before accessing hospital/ICU care. The discussion identifies the universal nature of the health system in Sweden but does not discuss

how decisions to allocate ICU beds and therefore facilities to ventilate may have been taken during the study period given the pressures that the health service was experiencing at the time. These factors mean that there are uncertainties around the outcome. An analysis of death following diagnosis of COVID-19, especially over the longer time period that has now elapsed since the original study, may provide more robust findings.

Author response: We would like to thank the reviewer both regarding the comment that our study is timely and make good use of Swedish national registers but also for many very valuable comments. We agree with the reviewer that using mechanical ventilation as an outcome may have some limitations as it does not address the outcomes of those patients where admission to ICU and/or mechanical ventilation was not considered appropriate, and/or of those who died before admission to the ICU. However, there are also advantages with this outcome by using a nationwide ICU-register. Thereby, we gain a large homogeneous study sample of patients who experienced a severe disease that is representative for the vast majority of the population in which admission to ICU in fact was considered appropriate. The considerations and decisions on how to allocate the ICU beds and subsequently mechanical ventilation cannot be found in the data sources that we have available. In general, the ICU capacity in Sweden was increased to a very high number (we never hit the upper limit) which, therefore, did not force the clinicians to prioritize as hard as first was assumed. Patients with extensive co-morbidities and high frailty score may not have been admitted to the ICU for mechanical ventilation if it was considered inappropriate but at large to a similar extent as during a normal period.

We agree with the reviewer that this information is of interest to the reader and it has been added in the limitations section of the revised version. In order to address this question further, we have performed a sensitivity analysis using all admissions to ICU (with or without mechanical ventilation) as an alternative outcome which is presented in the manuscript text and in detail in supplemental table 6 in the revised version. Confirmatory, in this analysis, the associations with cardiometabolic disease (the focus of the paper) were similar.

It also shows indeed that conditions such as heart failure, COPD and atrial fibrillation were overrepresented in the ICU population. This is now reported in the results section.

Unfortunately, the current study data base does not include data on mortality.

Overall, we acknowledge the important limitations mentioned by the reviewer and have added them to the revised version, but we do believe that the advantages with the outcome outweigh the possible limitations.

The paper repeatedly states that no other paper has assessed the risk associated with cardiometabolic risk factors and severe COVID-19 in comparison to a control population with matched demographic characteristics. However, there are two very large scale studies of English populations which use administrative data to consider this question

[\(https://eur01.safelinks.protection.outlook.com/?url=https%3A%2F%2Fpubmed.ncbi.nlm.nih.gov/33111111/\)](https://eur01.safelinks.protection.outlook.com/?url=https%3A%2F%2Fpubmed.ncbi.nlm.nih.gov/33111111/)

h.gov%2F32640463%2F&data=04%7C01%7CPer.Svensson%40ki.se%7Cfda9625e672f4c

19d07108d89b5faf4d%7Cbff7eef1cf4b4f32be3da1dda043c05d%7C0%7C0%7C63743018255

4317487%7CUnknown%7CTWFpbGZsb3d8eyJWIjoiMC4wLjAwMDAiLCJQIjoiV2luMzliLCJBTiI6

lk1haWwiLCJXVCI6Mn0%3D%7C1000&sdata=oOOdm6cfkpUT1cVK1Y19n0vfWj3l35OoR

jetPA%2B%2BVJc%3D&reserved=0 and

[<k1haWwiLCJXVCI6Mn0%3D%7C1000&sdata=b57XmeQHDFSXD6oGsIQzNCzzwLmTJKBUtTuBI2SzMpU%3D&reserved=0>\). Both of these papers use death related to COVID-19 as the outcome and review cardiometabolic factors in the context of social demographic characteristics. This paper needs to be reviewed after considering these papers and be considered as a study confirming these associations in a Swedish population.](https://eur01.safelinks.protection.outlook.com/?url=https%3A%2F%2Fpubmed.ncbi.nlm.nih.gov%2F32798472%2F&data=04%7C01%7CPer.Svensson%40ki.se%7Cda9625e672f4c19d07108d89b5faf4d%7Cbff7eef1cf4b4f32be3da1dda043c05d%7C0%7C0%7C637430182554317487%7CUnknown%7CTWFpbGZsb3d8eyJWlloiMC4wLjAwMDAiLCJQIjoiV2luMzliLCJBTiI6I<div data-bbox=)

We would like to thank the reviewer for mentioning of the two UK studies. We have now included both important references in the revised version of the manuscript as well as the accompanying study published simultaneously in Lancet Diabetes & Endocrinology on risk factors for death in COVID-19 in diabetic patients. This is a rapidly evolving field and, in particular, the Lancet D&E studies had not been published at the time of the first submission of our manuscript. We agree that we confirm the Barron et al findings for diabetes but also that we extend their findings with a different outcome and more detailed information on hypertension and some other important comorbidities as well as the use of individual level socioeconomic data (educational status). We have adhered to the reviewers' suggestion in the revision of the manuscript.

Some minor points for consideration

- **The dates of the study are not reported in the methods section.**

Author response: The dates of the study was between 1st of March until 11th of May 2020 and was reported in the beginning of the methods section. This information has now also been clarified in the outcome paragraph.

- **The time period in which co-morbidities were identified is only specified for some conditions.**

Author response: Thanks for this comment, the time frame for the collection of diagnoses was 15 years before admission. This information was specified in the "national registries and data collection" paragraph but have now been clarified also in the "definition of exposure" paragraph.

- **The methods section does not state how COVID-19 status was identified.**

Author response: The COVID-19 diagnosis was confirmed by a positive test upon admission to the ICU and reported to the SIR by the treating clinician or other personnel at the ICU. This information has been clarified.

- **The methods section does not include any details of the information governance/data protection processes that facilitated this work.**

Author response: Thanks for this question. In the revised version we present more detailed information on the laws and regulations that governs the Swedish patient register and NBHW (national board of health and welfare). This is now presented in the register information section in supplementary methods.

- **The results section could be improved by quoting the specific odds ratios and confidence intervals rather than relying on looking up all values in the tables.**

Author response: This is a good suggestion; we agree that this would improve readability. The results section has been revised and now incorporate the most important OR:s and CI:s

VERSION 2 – REVIEW

REVIEWER	Norbert Stefan University of Tübingen, Germany
REVIEW RETURNED	21-Dec-2020

GENERAL COMMENTS	The authors have very well addressed the critical points.
---

REVIEWER	Naomi Holman University of Glasgow, UK
REVIEW RETURNED	08-Jan-2021

GENERAL COMMENTS	Thank you for addressing the points raised by the reviewers.
--